# Wastewater Surveillance Captured an Increase in Adenovirus Circulation in Milan (Italy) during the First Quarter of 2022

**DOI:** 10.3390/v14112351

**Published:** 2022-10-26

**Authors:** Laura Pellegrinelli, Sara Colonia Uceda Renteria, Ferruccio Ceriotti, Emanuela Ammoni, Cristina Galli, Arlinda Seiti, Sara Castiglioni, Danilo Cereda, Sandro Binda, Elena Pariani

**Affiliations:** 1Department of Biomedical Sciences for Health, University of Milan, 20133 Milan, Italy; 2Fondazione IRCCS Ca’ Granda Ospedale Maggiore Policlinico, 20122 Milan, Italy; 3Directorate General for Health, Lombardy Region, 20124 Milan, Italy; 4Department of Environmental Sciences, Istituto di Ricerche Farmacologiche Mario Negri IRCCS, 20156 Milan, Italy

**Keywords:** wastewater surveillance, molecular characterization, adenovirus, adenovirus F-41, severe acute hepatitis of unknown etiology

## Abstract

The quantification and molecular characterization of the AdV genome in urban wastewater samples (WWSs) collected weekly at a wastewater treatment plant (WWTP) in Milan from 1 January 2021 (week 2021-01) to 1 May 2022 (week 2022-17) were performed. The concentration of the AdV genome was graphically compared with the AdV positive rate observed in the respiratory/gastrointestinal specimens from individuals hospitalized with acute respiratory/gastrointestinal infections collected from one of the major hospitals in Milan in the same time series. An increase in the AdV circulation in WWSs was seen from November 2021, peaking in March 2022 and overlapped with an increase in the AdV positive rate in respiratory/fecal samples from individuals hospitalized with acute respiratory/gastrointestinal infections. The molecular characterization of the hexon hypervariable region of loop 1 of AdV revealed the presence of the species F type 41 in WWSs collected from February 2022 to April 2022. The wastewater surveillance of AdV can provide crucial epidemiological characteristics regarding AdV, particularly where no clinical surveillance is ongoing. The increase in the AdV circulation in Milan both in WWSs and clinical samples temporally overlapped with the outbreak of severe acute pediatric hepatitis observed in Europe and needs to be better investigated.

## 1. Introduction

Currently (29 September 2022), at least 33 countries in five World Health Organization (WHO) regions have reported 555 probable cases of severe acute hepatitis of unknown etiology, according to the WHO case definition, in previously healthy children since October 2021 [1,2]; of these probable cases of severe acute hepatitis, 36 (6.8%) have been identified in Italy. The common causes of hepatitis have been ruled out [3], and adenovirus (AdV) has become one of the etiological candidates since laboratory investigations have ascertained that 77% of children with severe acute hepatitis in the UK tested positive to AdV detection, molecularly characterized as AdV species F type 41 (AdV-41F) [4]. Thus far, the US Centers for Disease Control and Prevention (CDC) have recommended to include AdV detection in laboratory testing of all suspected cases of acute hepatitis [5]. The role of SARS-CoV-2 infection is also under investigation: Brodin and Arditi have hypothesized the involvement of a superantigen-mediated immune activation as a consequence of AdV infection with intestinal tropism in children previously infected by SARS-CoV-2 and carrying viral reservoirs [6].

Wastewater surveillance can provide an early warning of viral spread in communities and give additional crucial information about virus circulation and prevalence, as recently demonstrated during the SARS-CoV-2 pandemic [7]. As people infected with AdV can shed the virus through feces [8], even if they do not have symptoms, the virus can then be detected in sewage, enabling wastewater surveillance to capture the presence and extent of AdV circulation in the community.

As the reference laboratory of environmental sewage surveillance of SARS-CoV-2 in the Lombardy region (Northern Italy) [9], we retrospectively detected the presence of the AdV genome in urban wastewater samples (WWSs) collected at one wastewater treatment plant (WWTP) in Milan from January 13, 2021 (week 2021-01) to May 1, 2022 (week 2022-17). The concentration of the AdV genome in the WWSs was graphically compared with the AdV positive rate observed in the respiratory/gastrointestinal specimens from individuals hospitalized with acute respiratory/gastrointestinal infections at the Fondazione IRCCS Ca’ Granda Ospedale Maggiore Policlinicoin Milan in the same time series.

## 2. Materials and Methods

### 2.1. AdV Genome Detection, Quantification and Molecular Characterization in Wastewater Samples

Overall, 52 composite 24 h raw WWSs collected within the framework of the national environmental surveillance of SARS-CoV-2 [8] at the inlet of “Milan San Rocco” WWTP (serving about 50% of the Milan population, with a catchment of 1,036,000 inhabitants) were retrospectively analyzed to detect AdV DNA. After viral concentration using PEG-8000-based centrifugation as previously described [10], DNA was extracted using QIAamp MinElute Virus Spin Kit (QIAGEN, Hilden, Germany), and AdV-DNA was detected and quantified by amplifying the AdV hexon gene (nt. 18,895–18,968) using an in-house real-time PCR assay [11]. Viral load was expressed as copies of genome per liter of WWS (cg/L) and normalized for daily wastewater flow by multiplying AdV-DNA concentration by the daily wastewater flow rate of the WWTP (m^3^/day). Normalized viral loads in WWSs were expressed as copies of genome per liter of WWS per day (cg/L/day).

Molecular characterization of AdV was performed using sequence analysis of the hexon hypervariable region (HVR) of loop 1 (nt. 131–331), as previously described by others [12].

### 2.2. Epidemiological Data

Real-life data were retrieved from the laboratory database of the molecular virology laboratory of the Fondazione IRCCS Ca’ Granda Ospedale Maggiore Policlinico in Milan (a research hospital with 36,000 hospitalizations/year) and analyzed to estimate the weekly AdV-positivity rate in the clinical samples routinely tested for AdV in the same time period. The queries used to extract the data were: i) request for AdV-DNA detection in respiratory/fecal specimens collected from patients of any age with acute respiratory/gastroenteric infections; and ii) respiratory/fecal samples collected from 1 January 2021 (week 2021-01) to May 1, 2022 (week 2022-17). The AdV-positivity rate was calculated by week, and it was expressed as crude proportion with corresponding 95% confidence interval (95% CI) calculated by the Mid-P exact test assuming a normal distribution.

Descriptive and quantitative analyses were conducted using Open Epi (version 3.01) and GraphPad Prism (version 9.3.1). Proportions were compared using the chi-square test based on binomial distribution. For continuous variables, the unpaired t-test was performed.

## 3. Results

Figure 1 represents a single time-series graph covering the time span from week 2021-01 to week 2022-17 comparing: (1) the load of AdV genome concentration in WWSs per day (cg/L/day); (2) the weekly AdV-positivity rate in the respiratory specimens; and (3) the weekly AdV-positivity rate in the fecal samples.

AdV-DNA was identified in 88.7% of the WWSs. In our samples series, the AdV load in the WWSs collected from week 01 to week 17 was statistically higher in 2022 than in 2021 (mean: 4.16 × 10^12^ cg/L/day vs. 1.92 × 10^14^ cg/L/day; *p* < 0.05). As displayed in Figure 1, the temporal analysis of the AdV-DNA load in our WWS series showed that the AdV-DNA concentration ranged between 5.97 × 10^11^ cg/L/day and 2.16 × 10^13^ cg/L/day up to week 2021-24; no AdV-DNA was identified in WWSs from week 2021-25 to week 2021-32; and the AdV-DNA concentration ranged between 5.97 × 10^12^ cg/L/day and 7.99 × 10^14^ cg/L/day from week 2021-33 to week 2022-17. The AdV-DNA load significantly increased after week 2021-44 (1.05 × 10^14^ cg/L/day; *p* < 0.05), reaching a peak in week 2022-10 (7.99 × 10^14^ cg/L/day) (Figure 1) (Appendix A).

Overall, 23 out of 44 (52.3%) AdV-positive WWSs had a viral concentration (>10^13^ cg/L/day) that allowed further sequencing analyses; a sequence result was obtained in 12 out of 23 (52.2%) wastewater samples. The sequence analysis was inconclusive in five samples as they revealed mixed electropherogram peaks, suggesting the co-presence of several AdV genotypes (the samples were collected before December 2021). A result was obtained for seven AdV-positive WWSs: one collected on 14 December 2021 was AdV A-12, and six WWSs collected between 16 February 2022 and 27 March 2022 were AdV F-41 (GenBank accession numbers: ON993776-ON993781).

A total of 3629 requests for AdV-DNA detection were recorded from week 2021-01 to week 2022-17: 2870 in respiratory samples and 759 in fecal samples. Considering the results of the respiratory samples, the overall AdV-positivity rate was 5.3% (110/2064; 95% CI: 4.4–6.3%) in 2021 and 6.1% (49/806; 95% CI: 4.7–7.9%) in 2022. To compare the same time span of 2021 and 2022, the first 17 weeks of each year were considered: the AdV-positivity rate was statistically higher (*p* = 0.017) in 2022 (49/806; 6.1%; 95% CI: 4.7–7.9%) than in 2021 (21/624; 3.4%; 95% CI: 2.2–5.1%).

The temporal analysis of AdV-positivity in the respiratory samples by week (Figure 1) shows two peaks in 2021, one in week 2021-27 (4/19; 21.1%; 95% CI: 8.5–43.3%) and one in week 2021-49 (10/44; 22.7%; 95% CI: 12.8–37%). The AdV-positivity rate in the respiratory specimens collected from week 2021-49 to week 2022-17 (73/1054; 6.9%; 95% CI: 5.5–8.6%) was statistically higher compared to that observed in samples collected in the previous period (86/1816; 4.7%; 3.8–5.8%). In 2022, the AdV-positivity rate reached a peak in week 2022-08 (5/34; 14.7%; 95% CI: 6.4–30%).

The overall AdV-positivity rate identified in the fecal samples collected from gastroenteric infection cases was 2% (10/494; 95% CI: 1.1–3.7%) in 2021 and 11.1% (28/252; 11.1%; 95% CI: 7.8–15.6%) in 2022. The temporal analysis of the AdV-positivity rate by week (Figure 1) shows no AdV detection in the fecal samples collected from week 2021-01 to week 2021-43 and an abrupt increase in the AdV-positivity rate from week 2021-43 with a peak of AdV-positivity in week 2021-45 (6/12; 50%; 95% CI: 25.4–74.6%) and in week 2022-11 (6/19; 31.6%; 95% CI: 15.4–54%).

As shown in Figure 1, the temporal trend of the AdV-positivity rate in fecal samples correlates better with data from WWSs than the AdV-positivity rate in respiratory samples.

## 4. Discussion

Wastewater surveillance has been proved to be a successful tool to monitor the circulation of pathogens in the general population, enabling the consideration of allinfected asymptomatic and symptomatic individuals shedding the virus through feces [7]. As AdV is eliminated by the fecal route [8] and considering that AdV is currently under investigation for its possible role in the development of severe acute hepatitis of unknown etiology [13,14], we assessed the circulation of AdV using a temporal analysis of the AdV-DNA concentration in a series of untreated urban WWSs collected weekly in a WWTP in Milan, and we matched the results with the AdV-positivity rates in clinical respiratory/fecal specimens in the same city.

To our knowledge, no papers on the application of wastewater-based epidemiology (WBE) to track AdV circulation have been published so far. Considering that no AdV surveillance systems are ongoing in Europe, the WBE approach we developed and implemented to track AdV circulation in the community can complement the epidemiological data on the circulation of this virus, thus adding novel epidemiological and virological information. In fact, WBE, first developed to monitor the circulation of polioviruses, is currently used worldwide as an early warning of poliovirus reintroduction; more recently, it has been implemented to control SARS-CoV-2 pandemic and to gain insights into its circulation [7,9,10]. The data from WBE might be useful to inform clinicians and public health practitioners on the epidemiology of circulating pathogens.

The results from the analysis of our WWSs series show that the AdV-DNA concentration increased from November 2021, reaching a peak on 9 March 2022. As the concentration of the AdV genome should be related to AdV shedding via the fecal route, its increasing trend may suggest a greater circulation of AdV in the community; this latter finding is supported by the real-life data retrospectively retrieved from the laboratory database of one of the major hospitals in Milan. In fact, the AdV-positivity rate in the respiratory/fecal samples collected from patients with acute respiratory/gastroenteric infections overlapped the curve of the AdV-DNA concentration in the sewage. This is more evident for the fecal samples data that better correlate with data from the WWSs. It is interesting to observe that AdV-positivity in the fecal samples peaked in mid-March (week 2022-11), three weeks later than the AdV-positivity peak in the respiratory samples, which occurred at the end of February (week 2022-08).

In our time series, the circulation of AdV was higher in the same period when pediatric cases of hepatitis without an apparent etiology in Europe were observed. As AdVs are a common cause of infection linked to mild-to-moderate gastrointestinal symptoms, it is puzzling to think that the AdValone can be the causative agent of the hepatitis upsurge in children [4].

Nearly half of AdV-positive WWSs had a viral load suitable for further sequencing analysis directly from the samples, avoiding strain selection by virus enrichment through cell culture. Unexpectedly, the AdV-positive WWS collected on 14 December 2021 was AdV species A type 12, whereas the WWSs collected from 16 February 2022 to 27 March 2022 were all AdV species F type 41 (AdV 41F). These genotypes were probably the predominant ones at the time of the sample collection and thus the prevalent circulating strains in the community. This increased circulation of AdV 41F from February 2022 to the end of March 2022 is in line with data from the UK that had identified the presence of AdV 41F in 77% of blood samples collected from children with acute hepatitis cases [4].

Unfortunately, the sequencing of the AdV-positive WWSs collected from January 2021 to November 2021 was inconclusive as the resulting sequences revealed mixed electropherogram peaks, thus suggesting the presence of multiple AdV genotypes that could not be discerned in the same samples.

Recently, it was hypothesized that an AdV infection may potentiate SARS-CoV-2 superantigen-mediated immune activation and cause the development of severe acute hepatitis [6,15]; following infection with SARS-CoV-2, viral reservoirs could, over time, lead to repeated superantigen-mediated immune cell activation, as shown in multisystem inflammatory syndrome [16]. If such viral reservoirs are present, and a child is subsequently infected with AdV, this superantigen-mediated effect could be much more pronounced and potentially give rise to immunopathology causing acute severe hepatitis [6].

This study has some limitations: no information on the AdV genotype in the clinical samples was available as this was not part of the routine testing; the genotyping data of the AdV in the WWSs are partial due to the complexity of these samples and the low viral concentration.

As it is difficult to conduct AdV virological surveillance based on clinical symptoms and considering the potential risk of AdV-related hepatitis which needs to be investigated, our work originally aimed at supplying relevant epidemiological and virological data on the circulation of AdV. Moreover, as no surveillance data on AdV are currently available, the implementation of laboratory-based surveillance for analyzing the virological data provided by laboratories that routinely perform molecular testing may enable the determination of baseline levels of viral circulation and any unexpected epidemiological change. Lastly, AdV sequencing by using a next-generation sequencing approach can help to obtain more detailed results to uncover cocirculating genotypes and/or to evaluate samples with a low viral load.

## 5. Conclusions

The continuous analysis of data coming from clinical settings coupled with the cutting-edge strategies of WBE will help supply relevant epidemiological and virological data on the circulation of AdV to increase current knowledge regarding the potential risk of human adenovirus-related diseases.

## Figures and Tables

**Figure 1 viruses-14-02351-f001:**
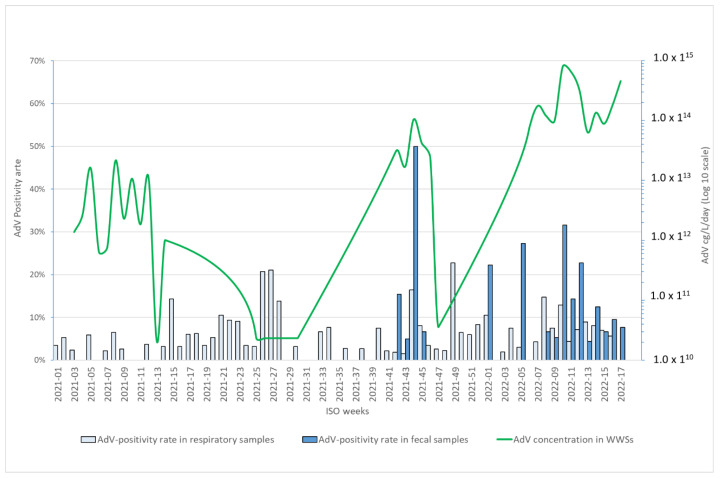
Quantification of AdV-DNA (cg/L/day) in urban wastewater samples collected in one wastewater treatment plant in Milan from week 2021-01 to week 2022-17 and the AdV-positivity rate by week in respiratory/fecal samples collected from inpatients with respiratory/gastroenteric infections at the Fondazione IRCCS Ca’ Granda Ospedale Maggiore Policlinico in Milan from week 2021-01 to week 2022-17.

## Data Availability

Data are available under request.

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
