# Peer review of "Wastewater Surveillance Captured an Increase in Adenovirus Circulation in Milan (Italy) during the First Quarter of 2022"

_viruses, 2022, doi:10.3390/v14112351_

Round 1

Reviewer 1 Report (Previous Reviewer 1)

The manuscript was improved following the corrections of the referees. My only criticism to the actual version is that data presented in Table 1is already present in figure, so this is unnecessary and could be inserted as a Suppl. file.

Author Response

We agree with the observation of Referee to insert Table 1 as supplementary material

Reviewer 2 Report (Previous Reviewer 2)

I do not recommend publication of this manuscript in Viruses since I believe the study is very limited in scope, highly specialized, and of limited interest to the general readership.

Author Response

To our knowledge - no papers on the application of wastewater-based epidemiology to track circulation of Adenovirus have been published so far. Thus, even though our study can be limited in scope, we believe that can give new insights on this topic thus adding novel information to the current knowledge and translation application. We have also disclosed and discussed all limitations of the study.

Round 2

Reviewer 2 Report (Previous Reviewer 2)

I accept the author's rebuttal about the significance of the research.

This manuscript is a resubmission of an earlier submission. The following is a list of the peer review reports and author responses from that submission.

Round 1

Reviewer 1 Report

The authors made substantial changes to a first version of the manuscript, it seems to me that the text now adequately addresses the possible limitations of the study in terms of establishing a definitive cause-and-effect relationship between a wave of HAdV41 and an eventual unequivocal relationship with acute hepatitis in children in the period. The text is more adequate and avoids inappropriate speculation in the present version, while still informing about the topic.

Author Response

We really thanks this referee for her/him positive feedback

Reviewer 2 Report

The authors present waster water surveillance of human adenovirus prevalence in Milan, Italy during the first quarter of 2022. The level of AdV genomes detected was compared with AdV-positivity rate in respiratory/ gastrointestinal specimens isolated from patients hospitalized with acute respiratory/gastrointestinal infection collected in a major hospital in the area during the same time period. An increase in AdV positivity rates was found during this time in both sets of samples. This correlated with AdV-41 in the waste water samples. The authors discuss these results in light of a recent pediatric hepatitis outbreak in the area.

I do not favor publication of this manuscript for several reasons. First, the study is very limited in scope (1 figure) and of limited interest to the general readership of Viruses. Second, there have been many articles speculating about the possible role of AdV-41 in recent pediatric hepatitis outbreaks. This manuscript does not extend these discussions in a meaningful way. Finally, recent reports have largely discounted the role of AdV-41 in these recent pediatric hepatitis cases.

Author Response

Reviewer ≠2

R2: The authors present waster water surveillance of human adenovirus prevalence in Milan. Italy during the first quarter of 2022. The level of AdV genomes detected was compared with AdV-positivity rate in respiratory/ gastrointestinal specimens isolated from patients hospitalized with acute respiratory/gastrointestinal infection collected in a major hospital in the area during the same time period. An increase in AdV positivity rates was found during this time in both sets of samples. This correlated with AdV-41 in the waste water samples. The authors discuss these results in light of a recent pediatric hepatitis outbreak in the area.

I do not favor publication of this manuscript for several reasons.

- First, the study is very limited in scope (1 figure) and of limited interest to the general readership of Viruses.

A2: To our knowledge, no papers on the application of wastewater-based epidemiology (WBE) to track the Adenovirus circulation have been published so far. In consideration that no Adenovirus (AdV) surveillance systems are currently ongoing in Europe, the WBE approach we developed and implemented to track AdV circulation in the general population could have the potential to complement epidemiological data on the circulation of this virus, thus adding novel information on the topic. Data from WBE can be useful to inform clinicians and public health practitioners to evaluate the circulation of viruses in the community. In fact, WBE - firstly developed to monitor polioviruses - it is now used worldwide as an early warning to uncover polioviruses circulation, and, more recently, WBE has been implemented to control SARS-CoV-2 and get insights in its circulation.

- Second, there have been many articles speculating about the possible role of AdV-41 in recent pediatric hepatitis outbreaks. This manuscript does not extend these discussions in a meaningful way.

A2: We agree that whereas there is an ever-increasing number of articles speculating about the possible role of AdV-41 in the recent pediatric hepatitis outbreaks, the role of the virus remains puzzling. As we stated in the discussion section (lines 171-175; lines 189-196; lines 201-207; lines 222-225), our work did not aim at establishing a cause-effect relationship between an increased circulation of AdV-F41 and an eventual relationship with acute hepatitis in children. We have intentionally avoided any inappropriate speculations in the text, in particular in the results and discussion sections, as it is clear in our minds that post-hoc does not mean propter hoc. As it is difficult

to conduct AdV virological surveillance based on clinical symptoms, and in consideration that the potential risk of AdV-related hepatitis needed to be investigated, our work originally aimed at supplying relevant epidemiological and virological data on AdV circulation in the community. Whether AdVs is involved in the hepatitis cases still requires further investigations, as endorsed by WHO, CDC and ECDC.

Reviewer 3 Report

In this article, the authors present data from wastewater surveillance of adenovirus in Milan over an 17 to 18 month period.  They compared the results with clinical hospital data.

The wastewater sampling and processing method is appropriate and the method they used to quantify virus is also appropriate.

That being said, the novelty of this work is very limited and does not add much knowledge to the field.  The only figure in the manuscript does not represent the wastewater data well as it does not show discretely where the data point are.  A table that contains all the wastewater data or putting points on the graph would be more useful than the continous line graph provided.  Furthermore, replicates of the measurement should be done with average and standard error demonstrated in order for more confidence in the trends to be had.

The sequencing attempts were not appropriate given the samples would represent a mixture of viruses and not single virus strains as you would typically find in a clinical sample.  For this reason, I am not surprised that the authors were not able to draw any conclusions from sequencing.  The authors should use high through amplicon sequencing of the variable hexon region in order to be able to comment on the identity and abundance of strains in wastewater.

Due to the limited novelty of the information provided in this manuscript, I cannot recommend the publication of this manuscript in its current form.

Author Response

Reviewer ≠3

R3: In this article. the authors present data from wastewater surveillance of adenovirus in Milan over an 17 to 18 month period.  They compared the results with clinical hospital data. The wastewater sampling and processing method is appropriate and the method they used to quantify virus is also appropriate.

That being said, the novelty of this work is very limited and does not add much knowledge to the field. 

A3. To our knowledge, no papers on the application of wastewater-based epidemiology (WBE) to track circulation of Adenovirus (AdV) have been published so far. The main strength of WBE is that it may allow to track the circulation of a pathogen that is eliminated by faces from symptomatic, paucisymptomatic  and asymptomatic infected individuals. This is the reason why WBE is not driven by clinical investigation based on clinical assessment. Moreover, data from WBE is not affected by test availability/indications and it is not restricted to specific groups of population (i.e age-group, symptoms) or settings (hospital setting or ambulatory care setting). One other powerful characteristic of WBE is that it can permit to construct  the temporal distribution of pathogen circulation, which is expected to be the highest during an epidemic affecting the population under investigation. In this light, WBE of AdV can works as an early warning of AdV circulation in the population also in absence of other epidemiological indicators.

R3: The only figure in the manuscript does not represent the wastewater data well as it does not show discretely where the data point. A table that contains all the wastewater data or putting points on the graph would be more useful than the continous line graph provided. Furthermore, replicates of the measurement should be done with average and standard error demonstrated in order for more confidence in the trends to be had.

A3: We thank the reviewer for her/his input. As suggested, we have added a table (Table 1) that reports Adenovirus positive rate (%) in respiratory samples and in gastrointestinal samples and Adenovirus RNA genome copies per 1 litre of wastewater samples per day. However, we believe that reporting average values and standard error of the replicates does not hinder different interpretation of the trend of viral circulation that the WBE approach attempts to do.

R3: The sequencing attempts were not appropriate given the samples would represent a mixture of viruses and not single virus strains as you would typically find in a clinical sample.  For this reason, I am not surprised that the authors were not able to draw any conclusions from sequencing.  The authors should use high through amplicon sequencing of the variable hexon region in order to be able to comment on the identity and abundance of strains in wastewater.

A3: We acknowledge that the Sanger sequencing might be not appropriate given the complex samples, as also discussed and highlighted as a main limitation of the study (lines 197-200). However, we were able to identify and characterize the presence of AdV-F41 in all wastewater samples collected from February 2022 to April 2022. Therefore, we believe that this finding can be representative of a wider circulation of AdV-F41 in the population, at least in the aforementioned period. We also believe that by sequencing the hexon hypervariable region (HVR) of loop 1 (nt. 131-331) of AdV in this study -as recommended by the CDC- instead of the variable hexon region can give enough information on the AdV genotypes circulation.
